# Calcium Channel Blockers and the Risk of Exacerbation in Patients with Chronic Obstructive Pulmonary Disease: A Nationwide Study of 48,488 Outpatients

**DOI:** 10.3390/biomedicines11071974

**Published:** 2023-07-12

**Authors:** Ema Rastoder, Pradeesh Sivapalan, Josefin Eklöf, Imane Achir Alispahic, Alexander Svorre Jordan, Christian B. Laursen, Jørgen Vestbo, Christine Jenkins, Rune Nielsen, Per Bakke, Gustavo Fernandez-Romero, Daniel Modin, Niklas Johansen, Filip Soeskov Davidovski, Tor Biering-Sørensen, Jørn Carlsen, Jens Ulrik Stæhr Jensen

**Affiliations:** 1Section of Respiratory Medicine, Herlev-Gentofte Hospital, 2900 Hellerup, Denmark; pradeesh.sivapalan.02@regionh.dk (P.S.); josefin.viktoria.ekloef@regionh.dk (J.E.); imane.achir.alispahic@regionh.dk (I.A.A.); alexander.svorre.jordan@regionh.dk (A.S.J.); jens.ulrik.jensen@regionh.dk (J.U.S.J.); 2Department of Respiratory Medicine, Odense University Hospital, 5000 Odense, Denmark; christian.b.laursen@rsyd.dk; 3Odense Respiratory Research Unit (ODIN), Department of Clinical Research, University of South Denmark, 5000 Odense, Denmark; 4Allergi-Og Lungeklinikken Vanløse, 2720 Copenhagen, Denmark; jorgen.vestbo@manchester.ac.uk; 5Division of Infection, Immunity and Respiratory Medicine, University of Manchester, Manchester M13 9MT, UK; 6Department of Thoracic Medicine, Concord Hospital, University of Sydney, Concord, NSW 2139, Australia; christine.jenkins@sydney.edu.au; 7Department of Clinical Science, University of Bergen, 5021 Bergen, Norway; rune.nielsen@uib.no; 8Department of Thoracic Medicine, Haukeland University Hospital, 5021 Bergen, Norway; per.bakke@uib.no; 9Department of Thoracic Medicine and Surgery, Lewis Katz School of Medicine at Temple University, Philadelphia, PA 19122, USA; gustavo.fernandezromero@tuhs.temple.edu; 10Section of Cardiovascular Medicine, Herlev-Gentofte Hospital, 2900 Hellerup, Denmark; danielmodin.md@gmail.com (D.M.); niklas.dyrby.johansen@regionh.dk (N.J.); filip.soeskov.davidovski@regionh.dk (F.S.D.); tor.biering@gmail.com (T.B.-S.); 11Department of Cardiology, Rigshospitalet, Copenhagen University Hospital, Department of Clinical Medicine, Faculty of Health and Medical Sciences, University of Copenhagen, 2200 Copenhagen, Denmark; joern.carlsen@regionh.dk; 12Department of Clinical Medicine, Faculty of Health and Medical Sciences, University of Copenhagen, 2200 Copenhagen, Denmark

**Keywords:** COPD, calcium channel blockers, thiazide, exacerbations, amlodipine, bendroflumethiazide

## Abstract

Patients with chronic obstructive pulmonary disease (COPD) are prone to developing arterial hypertension, and many patients are treated with the calcium channel blocker amlodipine. However, it remains unclear whether using this drug potentially affects the risk of acute severe exacerbations (AECOPD) and all-cause mortality in these patients. The data were collected from Danish national registries, containing complete information on health, prescriptions, hospital admissions, and outpatient clinic visits. The COPD patients (*n* = 48,488) were matched via propensity score on known predictors of the primary outcome in an active comparator design. One group was exposed to amlodipine treatment, and the other was exposed to bendroflumethiazide, since both of these drugs are considered to be the first choice for the treatment of arterial hypertension according to Danish guidelines. The use of amlodipine was associated with a reduced risk of death from all causes at the 1-year follow-up (hazard ratio 0.69, 95% confidence interval: 0.62–0.76) compared with the use of bendroflumethiazide in the matched patients. No difference in the risk of severe AECOPD was found. In the COPD patients, amlodipine use was associated with a lower risk of death from all causes compared with the use of bendroflumethiazide. Amlodipine seems to be a safe first choice for the treatment of arterial hypertension in COPD patients.

## 1. Introduction

Chronic obstructive pulmonary disease (COPD) caused 2.23 million deaths in 2019. This makes it the third leading cause of death globally [1,2]. COPD exacerbation (AECOPD) is the leading cause of mortality in this group of patients [3]. COPD is associated with a higher risk of comorbidities, affecting approximately 50% of patients; among these comorbidities is arterial hypertension [4]. International guidelines recommend the following: (i) thiazide diuretics, (ii) the calcium channel blocker amlodipine, (iii) ACE inhibitors, (iv) angiotensin II antagonists as first-line drugs [5]. Despite this recommendation, there is relatively little data regarding the use of antihypertensive drugs in COPD patients [6].

Beta-blockers are well investigated as antihypertensive treatment in COPD patients, and a multicenter randomized trial has shown a 91% increased risk of severe exacerbations in COPD patients who are treated with metoprolol [7]. However, data on the very commonly used dihydropyridine calcium channel blocker amlodipine are lacking [5,6,8]. Calcium channel blockers reduce the influx of Ca^2+^ to the smooth muscle cells of the arteries, leading to a relaxation of the smooth muscles and a decrease in blood pressure [9,10].

Studies with few participants have suggested that calcium channel blockers have an advantage of opposing contraction in the tracheobronchial smooth muscles, reinforcing the bronchodilator effect of ß_2_-agonists [6,11,12]. Calcium signaling plays a central role in the activation of inflammatory pathways. Activating calcium channels stimulate an influx of Ca^2+^ to the intracellular space in neutrophils, thereby activating the cells for an immune response [13,14]. Since calcium channels play an important part in the human innate immune response, it can be argued that calcium channel blockers could potentially decrease inflammation and alter the risk of COPD exacerbations.

The effect of amlodipine has never been systematically investigated in a large well-characterized population of COPD patients. The hypothesis of the current study is that COPD patients treated with amlodipine have a decreased risk of AECOPD hospitalization compared to COPD patients who are not treated with amlodipine.

## 2. Materials and Methods

### 2.1. Study Design

This is a nationwide observational cohort study. Patients with severe or very severe COPD were followed for 12 months. The period of inclusion was from 1 January 2010 to 19 January 2019. The index date was defined as the first amlodipine/bendroflumethiazide prescription after the first outpatient visit. For the patients who did not receive any of the two drugs, the index date was the first outpatient visit. Patients were followed for 12 months from the index date to the first occurrence of AECOPD, death from all causes, or until 19 January 2020. It was not possible to follow patients who migrated (143 individuals); therefore, they were excluded from the main and sensitivity analyses.

### 2.2. Study Sources

The following nationwide registers were used in this study:Danish nationwide register of outpatients with COPD [2] (DrCOPD—Danish Register of Chronic Obstructive Pulmonary Disease). The COPD diagnosis was based on spirometry and verified by a respiratory physician. From this register, we obtained the following variables: age, gender, forced expiratory volume in 1 s (FEV_1_), body mass index (BMI), date of out-patient visits, Medical Research Council (MRC) breathlessness score, smoking status, and the date of death.The Danish National Patient Registry (DNPR) [15], where all hospital admissions since 1995 are registered. Each hospital visit is coded by a physician with one primary diagnosis and one or more secondary diagnoses, according to the International Classification of Diseases, 10th revision (ICD10) from 1994. We extracted the following primary or secondary diagnoses: J19, J440–J441, DJ448-DJ449.The Danish National Health Service Prescription Database [16] (DNHSPD) holds all prescriptions that have been dispensed in Danish pharmacies since 2004, coded according to the Anatomical Therapeutic Chemical (ATC) classification system. The date of dispensation, the quantity dispensed, and the formulation and strength of the prescription are all included. Danish legislation requires pharmacies to provide information that ensures complete and accurate registration. The DNHSPD was used for information on amlodipine, bendroflumethiazide, prednisolone, and antibiotics.

### 2.3. Study Participants

DrCOPD was used to characterize Danish citizens with a COPD diagnosis between 1 January 2010 and 19 January 2019. These individuals were linked with the first amlodipine or bendroflumethiazide prescriptions (see Appendix A for ATC codes); this formed the base cohort. Patients with a cancer diagnosis were excluded from the study, as malignancy diagnosis may influence the outcome and the treatment (all diagnoses are presented by ICD-10 codes in Appendix B). Patients with carcinoma in situ were included. Patients receiving amlodipine and bendroflumethiazide concomitantly were excluded from the study, since the use of both treatments may reduce our ability to distinguish the effect of the medications. Missing values for FEV_1_ and BMI in a minority of patients were handled by using the last observation registered prior to the outpatient visit. In a few participants, we used the median value (imputation). This strategy was used to avoid selection bias. All comorbidities within ten years prior to the study entry were registered. Patients had to have at least one hospital contact where the comorbidity was registered as a primary or secondary diagnosis in the DNPR to be considered a participant with a comorbidity.

### 2.4. Outcomes

The following three outcomes were examined at 12 months after study entry: (i) severe exacerbations of COPD, defined as hospitalization-requiring COPD exacerbation; (ii) death from all causes; (iii) moderate exacerbations of COPD, defined as the patient receiving antibiotic treatment combined with prednisolone treatment, following the GOLD classification [17].

### 2.5. Statistics

Baseline characteristics were presented as frequencies and proportions for the active comparator and unmatched populations.

### 2.6. Main Analysis

As our main analysis, we performed an active comparator analysis between patients receiving amlodipine and patients receiving bendroflumethiazide. We chose bendroflumethiazide as the active comparator, as its primary indication—like amlodipine—is hypertension [5]. The purpose of selecting an active comparator is to mitigate confounding by indication and other unmeasured patient characteristics [18]. We matched the two groups using propensity score matching (using Greedy Match from the MayoClinic) [19]. The study population was matched based on age, sex, smoking status, GOLD stage, FEV_1_, and comorbidities. An unadjusted multivariable Cox proportional hazards model was performed on the matched population.

### 2.7. Sensitivity Analysis

A separate analysis was performed using the entire eligible unmatched population. The population was divided into amlodipine users and non-users (non-users being patients that were not exposed to amlodipine but could receive bendroflumethiazide). A multivariable Cox proportional hazards model was performed to compare the amlodipine users to the non-users. An adjusted multivariable Cox proportional hazards model was performed on this unmatched population; we adjusted for the same matched variables as in the primary analysis. Model control investigating the proportional hazards assumption and test for linearity was performed to validate the Cox proportional hazards regression.

Statistical analysis was performed using SAS statistical software version 9.4 (SAS Institute, Cary, NC, USA). R studio 1.4.1106 was used to make the illustrations presented in this article.

### 2.8. Ethics Statement

The study was approved by the Danish Data Protection Agency, approval number P-2019-831. In Denmark, retrospective use of register data does not require ethics approval or patient consent.

## 3. Results

### 3.1. Descriptive Analyses

Of the 64,380 outpatients registered in the DrCOPD between 1 January 2010 and 19 January 2019, 48,488 were eligible for this study (Figure 1). These patients were followed for 12 months from the first amlodipine/bendroflumethiazide prescription. The baseline characteristics of the patients in the unmatched population are presented in Table 1 for both the exposed and non-exposed group.

The unmatched population consists of 21,200 females and 19,060 males in the non-user group, and 4280 females and 3948 males in the group exposed to amlodipine. In general, the patients in the amlodipine group had more comorbidities than the control group. Other than that, the two groups were comparable (Table 1).

A total of 8852 patients received bendroflumethiazide, which is the drug that was used as the active comparator to amlodipine. Table 1 shows the baseline characteristics of both groups. In total, 8542 patients were matched on the variables described previously—4271 from each group. By our assessments, these groups were large enough for the active comparator to be our primary analysis.

### 3.2. Statistical Analyses

#### 3.2.1. Main Analysis

The group receiving bendroflumethiazide had 2849 events of severe exacerbations, 119 died, and a total of 2968 (69.5%) had an event of either severe AECOD or death within the 12-month follow-up. In the group receiving amlodipine, 2803 patients had an event of severe exacerbation of COPD and 59 died; 2862 (67%) had an event within the 12-month follow-up. There was no difference between the two groups in the risk of severe AECOPD (hazard ratio (HR): 1.05, confidence interval (CI) (1.00–1.11), *p*-value: 0.06) (Figure 2a). However, we found that amlodipine use was associated with a decreased risk of death from all causes within 12 months compared with bendroflumethiazide use (HR: 0.69, CI (0.62–0.76), *p*-value: <0.0001) (Figure 2b).

There was no difference between the two groups in the risk of moderate or severe AECOPD (HR: 1.04, CI (1.00–1.11), *p*-value: <0.05).

#### 3.2.2. Sensitivity Analysis

An adjusted Cox proportional hazards model was applied on the unmatched population at the 12-month follow-up. The Cox analysis was adjusted for the same variables that were used to match our primary analysis. Treatment with amlodipine was associated with a reduction in the risk of AECOPD (HR 0.88, 95% CI (0.81–0.95), *p*-value = 0.0013). The same was seen when looking at death (HR 0.78, 95% CI (0.69–0.87), *p*-value = <0.0001).

Two comparable groups were constructed using the propensity score matching of the whole population, with 4271 participants in each group. The propensity score matching analysis showed a similar pattern as the primary analysis, i.e., there was no significant difference between the two treatments when looking at severe AECOPD (HR: 1.05, CI (1.00–1.11), *p*-value: 0.049) but there was still a reduction in mortality (HR: 0.68, CI (0.61–0.75), *p*-value: <0.0001). When looking at severe and moderate AECOPD, there was no significant difference between the two groups (HR: 1.06, CI (1.01–1.12), *p*-value: <0.03).

## 4. Discussion

In this nationwide observational cohort study including all Danish COPD outpatients, a decreased risk of death from all causes was found in the individuals receiving amlodipine compared to the matched individuals using bendroflumethiazide, even though the population in the amlodipine group had a slightly higher prevalence of comorbidities such as diabetes, myocardial infarction, stroke, and peripheral vascular disease. When investigating the risk of severe or moderate COPD exacerbation, no difference was found between the two exposures.

The effect of amlodipine on the risk of severe or moderate COPD exacerbation, as well as all-cause mortality, has not previously been investigated systematically in a large, well-characterized population of COPD patients with a complete follow-up.

A multicenter randomized clinical trial (RCT) compared a combination treatment consisting of amlodipine and perindopril with a combination treatment consisting of atenolol and bendroflumethiazide and potassium, with the primary outcomes being non-fatal myocardial infarction and fatal coronary heart disease (CHD) [20]. The study showed a tendency of increased mortality and non-fatal myocardial infarction in the group receiving atenolol and bendroflumethiazide and potassium, though it was not significant [20]. Another RCT assigned participants with mild to moderate essential hypertension to one of the four groups, (i) bendroflumethiazide 1.25 mg/day and potassium chloride, (ii) bendroflumethiazide 2.5 mg/day and potassium chloride, (iii) amlodipine 5 mg/day, or (iv) enalapril 10 mg/day, to compare the efficacy and safety of treatments [21]. The trial found that all treatments reduced diastolic blood pressure; however, the reduction in the diastolic blood pressure on amlodipine was significantly higher than on the three other treatment combinations [21]. Both trials support the safety and efficacy of amlodipine use. A likelihood of better controlled blood pressure in patients treated with amlodipine may be one of the reasons why we see a lower mortality in this group of patients. It was suggested that amlodipine has an anti-inflammatory effect. The calcium channel blocker decreases Ca^2+^ entry into the cells by binding to voltage-gated Ca^2+^ channels (L, T, N, and P channels) [22]. An increase in extracellular Ca^2+^ may increase the stimulation of immune cells including neutrophiles [23]. When activated by the Ca^2+^ influx to the cells, the neutrophils emigrate to inflamed tissue [23].

Even though amlodipine is a frequently used treatment, and COPD is a very common disease, this is, to our knowledge, the first nationwide study to investigate the impact of amlodipine on severe and moderate exacerbation and all-cause mortality in patients with COPD. This study had a large population with over 40,000 outpatients. Since all patients were registered in DrCOPD, we were able to ensure a correct COPD diagnosis, which was specialist and spirometry verified. The population was well characterized with registrations of BMI, FEV_1_, age, smoking status, and comorbidities; due to this, we were able to adjust for confounders in our statistical models and thereby reduce the risk of bias. Furthermore, in Denmark, receiving amlodipine or bendroflumethiazide requires a prescription, and all prescriptions are registered in the national prescription database; this assured a high degree of data completeness. Additionally, hospital admissions and deaths are registered; in short, these events cannot happen in our country without being registered centrally. This secured the totality of data registration and follow-up. For our primary analysis, we used bendroflumethiazide as an active comparator, which reduces the risk of unmeasured “new user” bias. For the same reason, we propensity score matched the sensitivity analysis to several known predictors of the outcome, and the survival analysis was adjusted for the same variables. Both analyses seemed quite balanced on the key confounders.

Our study has limitations. Firstly, even though adjustments were made in the Cox analysis, confounding by indication cannot be completely ruled out. Secondly, the national prescription database makes it possible to see the prescriptions that a patient has received, but it contains no data on adherence. We can see if the prescriptions were collected, which makes it very likely that the treatment was used by the patient. Nevertheless, if non-adherence occurred, misclassification bias would be the consequence. Although this may be a challenge, the active comparator design assured that all patients did actually receive a treatment, and no plausible reason exists for imbalance in a possible non-adherence between the two groups.

## 5. Conclusions

In conclusion, we observed a lower risk of death from all causes in the COPD patients who used amlodipine than we did in the matched and comparable COPD patients who used another first-choice drug for arterial hypertension, namely, bendroflumethiazide. Thus, the use of amlodipine seems to be safe for COPD patients. Although our data are comforting regarding amlodipine use in COPD patients, they cannot lead to conclusions regarding causality based on the observational nature of the data. Since COPD is very prevalent worldwide, and since amlodipine is a frequently used first choice for arterial hypertension, we argue that a large-scale RCT should be performed to find out if this treatment is not only safe, but may actually benefit COPD patients.

## Figures and Tables

**Figure 1 biomedicines-11-01974-f001:**
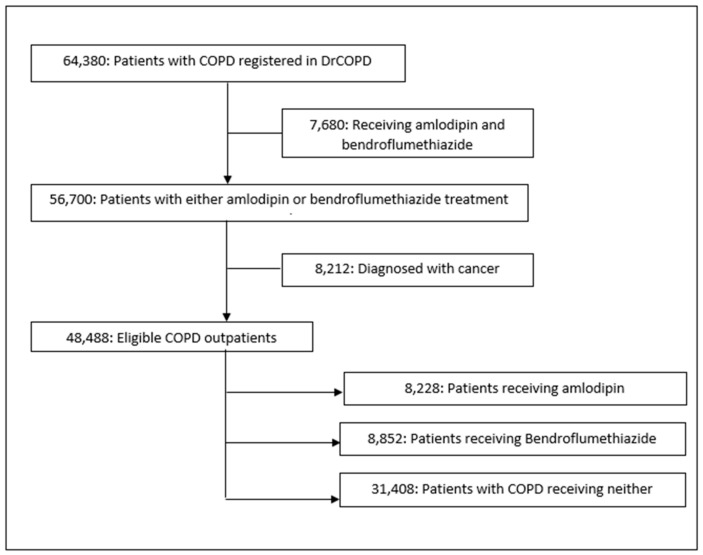
Study flowchart. DrCOPD, Danish Register of Chronic Obstructive Pulmonary Disease.

**Figure 2 biomedicines-11-01974-f002:**
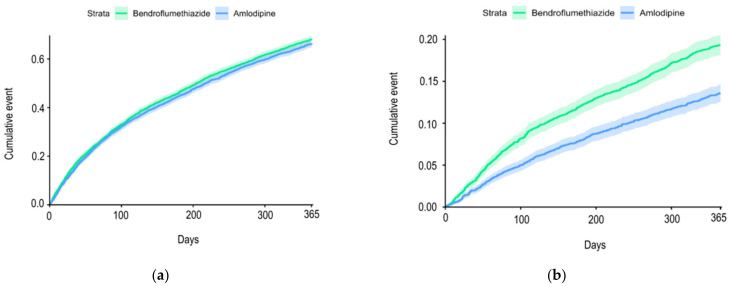
(**a**) Cumulative incidence curve for acute exacerbations from baseline until the first event within the 12-month follow-up period among chronic obstructive pulmonary disease outpatients treated with amlodipine vs. bendroflumethiazide; (**b**) cumulative incidence curve for all-cause mortality from baseline until the first event within the 12-month follow-up period among chronic obstructive pulmonary disease outpatients treated with amlodipine vs. bendroflumethiazide.

**Table 1 biomedicines-11-01974-t001:** The demographic and clinical characteristics for outpatients with COPD in the period from 1 January 2010 to 19 January 2019.

	Active Comparator Population		Entire Unmatched Population	
	Amlodipine Users	Bendroflumethiazide Users	Amlodipine Users	Non-Users
	(*n* = 4271)	(*n* = 4271)	(*n* = 8228)	(*n* = 40,260)
Age, years, median (IQR)	70 (63–77)	72 (64–78)	71 (64–78)	70 (62–78)
Age				
≤62 years	986 (23.1%)	915 (21.4%)	1672 (20.3%)	10,819(26.9%)
63–70 years	1174 (27.5%)	1069 (25.0%)	2251 (27.4%)	9909 (24.6%)
71–77 years	1083 (25.4%)	1099 (25.7%)	2130 (25.9%)	9353 (23.2%)
≥78 years	1028 (24.1%)	1188 (27.8%)	2175 (26.4%)	10,179 (25.3%)
Sex				
Female	2213 (51.8%)	2182 (51.1%)	4280 (52.0%)	21,200 (52.7%)
Male	2058 (48.2%)	2089 (48.9%)	3948 (48.0%)	19,060 (47.3%)
Smoking				
Active	1508 (35.3%)	1255 (29.4%)	2718 (33.0%)	13,826 (34.3%)
Former smoker/never smoked	2394 (56.1%)	2648 (62.0%)	4874 (59.2%)	22,626 (56.2%)
Missing data	372 (8.7%)	368 (8.6%)	636 (7.73%)	3808 (9.5%)
BMI				
≤18.4 kg/m^2^	268 (6.3%)	318 (7.5%)	558 (6.8%)	3755 (9.3%)
18.5–24.9 kg/m^2^	1452 (34.0%)	1303 (30.5%)	2783 (33.8%)	14,304 (35.5%)
25–30 kg/m^2^	1700 (39.8%)	1757 (41.1%)	3194 (38.8%)	15,699 (38.8%)
>30 kg/m^2^	851 (19.9%)	893 (20.9%)	1693 (20.6%)	6501 (16.2%)
Missing	0 (0.0%)	0 (0.0%)	0 (0.0%)	1 (0.0%)
GOLD stage				
GOLD 1	299 (7.0%)	231 (5.4%)	496 (6.0%)	2386 (5.9%)
GOLD 2	2302 (53.9%)	1994 (46.7%)	4059 (49.3%)	18,364 (45.6%)
GOLD 3	1352 (31.7%)	1485 (34.8%)	2835 (34.5%)	13,889 (34.5%)
GOLD 4	318 (7.5%)	561 (13.1%)	838 (10.2%)	5621 (14.0%)
Severe exacerbations 12 months prior to baseline				
0	2937 (68.8%)	2921 (68.4%)	5300 (64.4%)	26,599 (66.1%)
1	259 (6.1%)	257 (6.0%)	484 (5.9%)	2346 (5.8%)
≥2	1075 (25.2%)	1093 (25.6%)	2444 (29.7%)	11,315(28.1%)
Comorbidities				
Diabetes mellitus	333 (7.8%)	238 (5.6%)	801 (9.7%)	2187 (5.4%)
Myocardial infarction	342 (8.0%)	226 (5.3%)	766 (9.3%)	2391 (5.9%)
Peripheral vascular disease	534 (12.5%)	373 (8.7%)	1204 (14.6%)	3446 (8.6%)
Cerebrovascular disease	417 (9.8%)	349 (8.2%)	850 (10.3%)	2888 (7.2%)
Renal failure	71 (1.7%)	61 (1.4%)	622 (7.6%)	1020 (2.5%)
Heart failure	276 (6.5%)	280 (6.6%)	869 (10.6%)	3954 (9.8%)
Depression	17 (0.4%)	11 (0.3%)	52 (0.6%)	198 (0.5%)
Asthma	268 (6.3%)	314 (7.4%)	532 (6.5%)	2970 (7.4%)
Atrial fibrillation	375 (8.8%)	437 (10.2%)	1019 (12.4%)	4376 (10.9%)

## Data Availability

We believe that knowledge sharing increases the quality and quantity of scientific results. The sharing of relevant data will be discussed within the study group upon reasonable request.

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
