# Peer review of "Calcium Channel Blockers and the Risk of Exacerbation in Patients with Chronic Obstructive Pulmonary Disease: A Nationwide Study of 48,488 Outpatients"

_biomedicines, 2023, doi:10.3390/biomedicines11071974_

Round 1

Reviewer 1 Report

This is an interesting article that could be accepted after a monor revision.

The following questions should be addressed:

1) It is well-known that L-type voltage-gated potassium channels are also present in heart muscle. Blockage of these channels by beta blockers may slow down the heart rate and decrease of the power of the heart contraction. Is there any evidence about the influence of amlodipine on the heart action, which may be harmful especially in case of patients with cardiac problems ?

2) I could not see the data obtained for the sensitivity analysis (point 3.2.2) I think it will be worthy to show them.

3) I could not see the Figure 3, instead I saw the Figures 2A and 2B.

4) I would suggest a moderate correction of English used. 

 I would suggest a moderate correction of English used.  Please, read the whole text carefully with an aid of an English native speaker. 

Author Response

  1. It is well-known that L-type voltage-gated potassium channels are also present in heart muscle. Blockage of these channels by beta blockers may slow down the heart rate and decrease of the power of the heart contraction. Is there any evidence about the influence of amlodipine on the heart action, which may be harmful especially in case of patients with cardiac problems?

Thank you for this interesting question. The effect of amlodipine on the heart is very relevant, but it deviates from hypotheses and our aim of this article, this is why we haven’t focused on that interaction.

  1. I could not see the data obtained for the sensitivity analysis (point 3.2.2) I think it will be worthy to show them.

The data is listed in table 1 under entire unmatched population.

  1. I could not see the Figure 3, instead I saw the Figures 2A and 2B.

Thank you for bringing this to our attention, it has now been corrected.

  1. I would suggest a moderate correction of English used. 

Thank you, this has now been done.

Reviewer 2 Report

The manuscript gives information about the impact of amlodipine in de COPD. The work is significant as the number of patients is high

The structure is correct and the methodology is of interest as it can be used for other drugs

Author Response

The manuscript gives information about the impact of amlodipine in de COPD. The work is significant as the number of patients is high

The structure is correct and the methodology is of interest as it can be used for other drugs.

Thank you very much. We also believe that the results are interesting and important.

Reviewer 3 Report

1. Figure 1 shows the Study flowchart. However, another group of 4,271 is discussed twice in the text. Why not add all groups to a single scheme so that there are no questions?

2. Under figure 2, provide a table with the data.

Author Response

  1. Figure 1 shows the Study flowchart. However, another group of 4,271 is discussed twice in the text. Why not add all groups to a single scheme so that there are no questions?

Thank you for this question. The flowchart shows the whole eligible population divided into amlodipine users, bendroflumethiazide users and non-users. That is the population before any analysis are done. The two groups of 4,271 are obtained after we propensity match the amlodipine and the bendroflumethiazide users. We haven’t included it in the flow chart since the chart is just raw data.  

  1. Under figure 2, provide a table with the data.

Thank you for the comment. The data from for figure 2 is seen in table 1, where the matched amlodipine and bendroflumethiazide is shown.